# Microbial Diversity of Terrestrial Geothermal Springs in Armenia and Nagorno-Karabakh: A Review

**DOI:** 10.3390/microorganisms9071473

**Published:** 2021-07-09

**Authors:** Ani Saghatelyan, Armine Margaryan, Hovik Panosyan, Nils-Kåre Birkeland

**Affiliations:** 1Department of Biochemistry, Microbiology and Biotechnology, Yerevan State University, Alex Manoogian 1, Yerevan 0025, Armenia; ani.saghatelyan@gmail.com (A.S.); arminemargaryan@ysu.am (A.M.); 2Research Institute of Biology, Yerevan State University, Alex Manoogian 1, Yerevan 0025, Armenia; 3Department of Biological Sciences, University of Bergen, P.O. Box 7803, NO-5020 Bergen, Norway

**Keywords:** geothermal springs, microbial diversity, thermophiles, culture-dependent and -independent techniques, biotechnology, thermozymes, bioactive compounds

## Abstract

The microbial diversity of high-altitude geothermal springs has been recently assessed to explore their biotechnological potential. However, little is known regarding the microbiota of similar ecosystems located on the Armenian Highland. This review summarizes the known information on the microbiota of nine high-altitude mineralized geothermal springs (temperature range 25.8–70 °C and pH range 6.0–7.5) in Armenia and Nagorno-Karabakh. All these geothermal springs are at altitudes ranging from 960–2090 m above sea level and are located on the Alpide (Alpine–Himalayan) orogenic belt, a seismically active region. A mixed-cation mixed-anion composition, with total mineralization of 0.5 mg/L, has been identified for these thermal springs. The taxonomic diversity of hot spring microbiomes has been examined using culture-independent approaches, including denaturing gradient gel electrophoresis (DGGE), 16S rRNA gene library construction, 454 pyrosequencing, and Illumina HiSeq. The bacterial phyla Proteobacteria, Bacteroidetes, Cyanobacteria, and Firmicutes are the predominant life forms in the studied springs. Archaea mainly include the phyla Euryarchaeota, Crenarchaeota, and Thaumarchaeota, and comprise less than 1% of the prokaryotic community. Comparison of microbial diversity in springs from Karvachar with that described for other terrestrial hot springs revealed that Proteobacteria, Bacteroidetes, Actinobacteria, and Deinococcus–Thermus are the common bacterial groups in terrestrial hot springs. Contemporaneously, specific bacterial and archaeal taxa were observed in different springs. Evaluation of the carbon, sulfur, and nitrogen metabolism in these hot spring communities has revealed diversity in terms of metabolic activity. Temperature seems to be an important factor in shaping the microbial communities of these springs. Overall, the diversity and richness of the microbiota are negatively affected by increasing temperature. Other abiotic factors, including pH, mineralization, and geological history, also impact the structure and function of the microbial community. More than 130 bacterial and archaeal strains (*Bacillus*, *Geobacillus, Parageobacillus, Anoxybacillus, Paenibacillus, Brevibacillus Aeribacillus, Ureibacillus, Thermoactinomyces, Sporosarcina, Thermus, Rhodobacter, Thiospirillum, Thiocapsa, Rhodopseudomonas, Methylocaldum, Desulfomicrobium, Desulfovibrio, Treponema, Arcobacter, Nitropspira*, and *Methanoculleus*) have been reported, some of which may be representative of novel species (sharing 91–97% sequence identity with their closest matches in GenBank) and producers of thermozymes and biomolecules with potential biotechnological applications. Whole-genome shotgun sequencing of *T. scotoductus* K1, as well as of the potentially new *Treponema* sp. J25 and *Anoxybacillus* sp. K1, were performed. Most of the phyla identified by 16S rRNA were also identified using metagenomic approaches. Detailed characterization of thermophilic isolates indicate the potential of the studied springs as a source of biotechnologically valuable microbes and biomolecules.

## 1. Introduction

The natural geothermal springs found on our planet are located in tectonically active areas characterized by a relatively thin layer of the Earth’s outermost shell. Geothermal springs harbor enchanting microbes with unusual properties that can withstand harsh environmental conditions [1,2,3,4]. As analogs for the primitive Earth and models of ancient life, hot springs have become irreplaceable and attractive topics for research [5]. Studies on thermophiles provide insights into the origin of life on Earth and an understanding of the temperature range that defines the limits of life. Thermophiles are particularly important in astrobiology [6] and biotechnology. 

Recently, scientists have focused on the microbial communities in extreme niches. Culture-based approaches were initially used to study microbial diversity [7]. Microbial exploration due to the development of molecular biological techniques has greatly improved in the recent decades. In addition to the classical microbiological methods, application of culture-independent approaches has considerably increased our understanding of the taxonomic and functional composition of the microbiota in hot springs [3,7]. This methodological improvement has provided more complete and versatile insights into the microbial communities of geothermal springs and their taxonomic diversity, adaptation mechanisms, and functional and ecological roles. However, cultivation-dependent methods are still required to isolate and phenotypically characterize pure cultures/new species and to evaluate their biotechnological potential. The combination of traditional and new molecular approaches has been extensively used to study the microbiota of geothermal springs in all continents, including Eurasia [8,9,10,11,12,13,14,15,16,17,18], North and South America [19,20], Africa [21,22], Australia, Oceania [23,24], and Antarctica [25].

Hot springs have been recognized as an important source of beneficial thermophiles belonging to the Bacteria and Archaea domains, which can produce valuable biomolecules and thermozymes. Owing to their thermostability and ability to withstand the high-temperature conditions often utilized in industrial practices, thermozymes have several industrial applications [26,27]. The use of thermophiles and their products in harsh industrial conditions at elevated temperatures has several advantages, including reduced risk of microbial contamination, lower viscosity of the reaction medium and increased mass transfer, diffusion coefficient, and solubility of substrates [2,4,5].

The geothermal high-altitude springs located in the Lesser Caucasus Mountains are natural reservoirs of as yet undescribed biotechnological resources of thermophiles. The geology of this region is complex and is conditioned by ongoing tectonic activity and volcanism [28,29]. GeothermEx investigated the geothermal resources of Armenia and identified numerous low-temperature resource areas (cooler than 100 °C) within regions of younger volcanic activity. Some of the studied springs have been identified as prospective sources of geothermal energy and are commercially usable for agriculture, space heating, hot water supply, and tourism [28].

Despite the wide distribution of geothermal springs throughout Armenia, the microbiological analysis of these hot springs has been limited. 

Therefore, this review summarizes the investigations of thermophilic microbial communities of nine geothermal springs located in the Armenian Highland (Armenia and Nagorno-Karabakh), focusing on their biology and biotechnological potency.

## 2. Physiochemical Profiling

Armenia and Nagorno-Karabakh are located in the Armenian Highlands of Western Asia and the Lesser Caucasus Mountains (Figure 1). The Caucasus Mountains are part of the Alpine–Himalayan orogenic belt, a seismically active region next to the circum-Pacific belt. The Caucasus Mountains were formed as a result of the collision between the Arabian and Eurasian tectonic plates. This caused an uplift and Cenozoic volcanic activity in the Lesser Caucasus Mountains [30]. Thus, the geology of this region is complex and is regularly subjected to strong earthquakes [31]. Tectonic activity and volcanism have been occurring continuously in this region since the Lower Pliocene or Upper Miocene period.

Nagorno-Karabakh is a mountainous region located on a major part of the Kuro-Araks lowland. It includes the eastern part of the Karabakh Plateau and the seismically active Artsakh valley.

Many geothermal springs are found in the territory of Armenia and Nagorno-Karabakh and geotectonic processes are still ongoing [32].

In total, nine high-altitude mineralized geothermal springs (with temperatures ranging from 25.8 to 70 °C) have been investigated. The studied geothermal springs are at high elevations, at altitudes ranging from 960–2090 m above sea level. Most of the studied springs were moderately hot or mesothermal springs with temperatures ranging from 37–50 °C. The lowest temperature was found in the Uyts warm spring (25.8 °C). The highest temperature was recorded in the hot springs at Karvachar (70 °C). All studied Armenian and Nagorno-Karabakhian hot springs have naturally circumneutral pH (6.0–7.5).

The chemical composition of spring water depends on the water–rock interactions occurring within the reservoir along the ascent path. All investigated Armenian and Nagorno-Karabakhian thermal springs showed mixed-cation mixed-anion compositions. The total mineralization was 0.5 mg/L, but was occasionally higher. Geothermal springs with relatively high temperatures are characterized by higher ratios of cations (sodium and potassium to calcium and magnesium) and anions (chloride to bicarbonate or sulfate to bicarbonate). Warm geothermal spring waters are characterized by high bivalent cations (Ca + Mg) and bicarbonate contents [28,32].

Ionic coupled plasma optical emission spectrometry, mass spectrometry, and ion chromatography were used to analyze the major and minor elements and anions in Arzakan geothermal spring water samples. Major and minor elements such as Na, K, Mg, Ca, Si, B, Sr, Li, Mn, Fe, and Ba have been found in these samples. The highest concentration was observed for sodium (1183 ppm) and the lowest concentration was observed for Ba (0.09 ppm). Analyses of anions in the water revealed the presence of Cl^−^ (297 ppm) and SO_4_^2−^ (200 ppm). The trace elements Cr, Co, Cu, and Zn were found to have concentrations (in ppb) ranging from 0.28 to 6.73 [33]. Hydrogen sulfide (H_2_S) and other gas bubbles of unknown compositions were also detected. Some of the studied springs have been used by local people and tourists as baths and spas. Balneological centers have been created in Arzakan and Jermuk, where the water and sediments of geothermal springs are widely used for therapeutic properties [32]. The detailed information regarding the location and physicochemical profiles of the Armenian and Nagorno-Karabakhian geothermal springs is summarized in Table 1.

## 3. Microbiological Analysis

The hot springs in Armenia and Nagorno Karabakh have been studied for different purposes: isolation and description of novel thermophilic strains or potentially new species; screening of different bio-resources; metagenomic investigation of hot spring microbiomes; and whole-genome analysis of prospective isolates for elucidating genes linked to phenotypic expression.

### 3.1. Cultivation-Dependent Studies

Several cultivation-based studies have been performed to describe novel strains/species (including their identification based on whole-genome analysis) and to characterize different bio-resources obtained from Armenian geothermal springs. Thermoenzyme producers, such as lipase, protease, amylase, and DNA polymerase producers, as well as exopolysaccharide (EPS) producers, have been isolated from targeted geothermal springs.

In total, more than 130 strains of 40 distinct species belonging to 22 different genera, namely *Bacillus, Geobacillus, Parageobacillus, Anoxybacillus, Paenibacillus, Brevibacillus Aeribacillus, Ureibacillus, Thermoactinomyces, Sporosarcina, Thermus, Rhodobacter, Thiospirillum, Thiocapsa, Rhodopseudomonas, Methylocaldum, Desulfomicrobium, Desulfovibrio, Treponema, Arcobacter, Nitrospira*, and *Methanoculleus*, have been isolated from the hot springs in Armenia and Nagorno-Karabakh. Among these isolates, many prospective strains have been selected for future studies to explore their potential in white (industrial), green (agricultural), and red (pharmaceutical) biotechnology [33,34,35,36,37,38,39,40,41,42,43,44,45,46,47,48,49,50,51,52,53,54].

All isolated bacilli (107 strains in total) were phylogenetically profiled based on their 16S rRNA gene sequences. More than 22 distinct species belonging to the genera *Aeribacillus, Anoxybacillus, Bacillus, Brevibacillus, Geobacillus, Parageobacillus, Paenibacillus,* and *Ureibacillus* were identified. Representatives of the genera *Bacillus, Geobacillus/Parageobacillus*, and *Anoxybacillus* were the most abundant. Many isolated bacilli shared 91–97% sequence identity with their closest matches in GenBank, indicating that they belonged to novel taxa, at least at the species level. All isolated thermophilic bacilli were tested for the production of enzymes such as lipases, proteases, and amylases, and several biotechnologically valuable enzyme producers were selected. Of these, 71% of the isolates were found to actively produce at least one or more extracellular protease, amylase, or lipase [34,35].

A thermophilic amylase-producing an *Anoxybacillus* sp. strain designated as K1 was isolated from sediment samples collected from a Karvachar hot spring. This strain is a moderately thermophilic facultative anaerobe with a growth temperature range of 45–70 °C (T_opt_ 60–65 °C). The growth of strain K1 was observed at pH 6–11 (pH_opt_ 8–9). Strain K1 (MK418417) was found to share 99% 16S rRNA sequence similarity and a genome average nucleotide identity (ANI) value of 94.5% with its closest relative, *Anoxybacillus flavithermus* DSM 2641^T^, suggesting that it represents a separate and novel species [36,37]. The strain *Anoxybacillus* sp. K1 (MK418417) was deposited in the German Collection of Microorganisms and Cell Cultures (DSMZ) under the number DSM106524.

Another thermophilic *Anoxybacillus* strain, designated as *Anoxybacillus* sp. LF_2, which can use dextrin and lactate as carbon sources, was isolated from the Jermuk geothermal spring. The partial 16S rRNA gene sequence analysis of the isolate confirmed 99% sequence similarity affiliation with *Anoxybacillus mongoliensis* [38]. The strain *Anoxybacillus* sp. LF_2 (KX018621) was deposited in the DSMZ under the number DSM101951.

Three *Bacillus* strains designated as *Bacillus*
*licheniformis* Akhurik 107 (KY203975), *Geobacillus* sp. Tatev 4 (KY203974), and Karvachar QB2 (KY203976) were isolated and characterized as active lipase producers. The lipase activities of the *B. licheniformis* Akhurik 107 and *Geobacillus* sp. Tatev 4 strains at the optimal pH (6–7) and temperature (55 °C, 65 °C) conditions were 0.89 U/mL and 3.4 U/mL, respectively. Primer sets were designed to sequence lipase genes from lipase producers. The studied lipases were confirmed to belong the lipase family I and GDSL esterase family II based on the conserved region analysis of lipase protein sequences and amino acid composition analysis. Zn^2+^ and Ca^2+^ have been shown to bind lipases [39].

The strains Akhurik 107, Tatev 4, and Karvachar QB2 were deposited at the Microbial Depository Center of Armenia (MDCA) under accession numbers MDC11855, MDC11856, and MDC11857, respectively.

Two other thermophilic lipase-producing bacilli strains, *Geobacillus* (*Parageobacillus*) *toebii* Tatev 5 and Tatev 6, were isolated from the Tatev hot spring. The lipase activity of these isolates at the optimal growth temperature and pH (65 °C, pH 7) was 70.3 and 80.7 U/mL, respectively [40].

Panosyan et al. [41] isolated and studied *Geobacillus thermodenitrificans* ArzA-6 and *Geobacillus* (*Parageobacillus*) *toebii* ArzA-8 as EPS-producing strains. The time and temperature of cultivation, as well as culture medium composition, were found to be the key factors for EPS production. The specific EPS production yield, calculated after 24 h of cultivation at 65 °C and pH 7.0 with fructose as the sole carbon source, was 0.27 g g^−1^ dry cells for *G. thermodenitrificans* ArzA-6 and 0.22 g g^−1^ dry cells for *G. toebii* ArzA-8. The molecular masses of the heteropolymeric EPSs were 5 × 10^5^ Da and 6 × 10^5^ Da for *G. thermodenitrificans* ArzA-6 and *P. toebii* ArzA-8, respectively. GC-MS, HPAE-PAD, and NMR analyses indicated that mannose was the major monomer of the biopolymers studied. The accession numbers of the strains deposited at the MDCA were MDC11858 and MDC11859 for ArzA-6 and ArzA-8, respectively.

A thermophilic spirochete *Treponema* sp. J25 was isolated from Jermuk geothermal spring. It was strictly anaerobic and could ferment xylan. Its optimal growth temperature was 55 °C at circumneutral pH. It showed only 95.1% 16S rRNA sequence similarity with *Treponema caldarium* H1^T^, indicating that strain J25 potentially represents a novel species in the genus *Treponema* [38]. DSM100394 is the accession number of *Treponema* sp. J25 (MG970326) in the DSMZ.

A novel strain of *Thermus scotoductus* (designated K1) from sludge samples of Karvachar hot spring was recently isolated [42]. The strain K1 was catalase- and oxidase-positive, with T_opt_ 65 °C and pH _opt_ 8. It could reduce nitrate in the anaerobic respiration process. *T. scotoductus* strain SA-01, recovered from a deep gold mine in South Africa, was found to be the closest relative species of strain K1, with 80% genome sequence similarity. The DNA polymerase of strain K1 (*Ts*K1) has been cloned and subsequently expressed in *Escherichia coli* host cells [43]. The purified enzyme could efficiently amplify 2.5 kb DNA products under the optimal reaction conditions (74 °C and pH 9), and required 3–5 mM Mg^2+^ for optimal activity. *Ts*K1 polymerase was stable for at least 1 h at 80 °C, with half-lives of 30 and 15 min at 88 °C and 95 °C, respectively [43].

Two aerobic strains, AkhA-12 (MK418253) and Tatev 35a (MK418408) (with growth T*_opt_* 50–55 °C and pH*_opt_* 7.0–7.4), were isolated and identified as representatives of the genus *Thermoactinomyces* from Akhurik and Tatev, respectively. Both strains were producers of extracellular hydrolases (amylase, lipase, and protease) [44].

A gammaproteobacterial methanotroph, *Methylocaldum* sp. AK-K6 (KP272135) was isolated by Islam et al. [45] from the Akhurik geothermal spring. The strain had a growth temperature range of 8–35 °C (T*opt* 25–28 °C) and pH range of 5.0–7.5 (pH_opt_ 6.4–7.0). The strains possessed type I intracytoplasmic membranes; based on their 16S rRNA gene sequences, they formed a separate clade in the family *Methylococcaceae*.

Another methanotrophic isolate, *Methylocaldum* sp. Arz-AM-1 (JQ929024), and an epsilonproteobacterial anaerobic chemoorganoheterotrophic isolate, *Arcobacter* sp. Arz-ANA-2 (JQ929025), were isolated from the Arzakan geothermal spring [33].

Two lactate-oxidizing and sulfate-reducing thermophilic bacteria, *Desulfomicrobium thermophilum* SRB_21 and *Desulfovibrio psychrotolerans* SRB_141 (T_opt_ 55 °C), were isolated from Jermuk hot spring. Strain SRB_21 (KX018622) could also grow lithotrophically with hydrogen as an electron donor [38,46].

Sediment samples from Arzakan and Jermuk geothermal springs were used to enrich nitrite-oxidizing bacteria (NOB) at 45–50 °C. *Nitrospira calida* and *Nitrospira moscoviensis* were the dominant species [47].

Acetoclastic and hydrogenotrophic methanogenic enrichment at 45 °C and 55 °C using sediments from Jermuk and Arzakan geothermal springs were obtained by Hedlund et al. [48].

Panosyan and Birkeland [33] reported the isolation of the archaeal methanogenic strain, *Methanoculleus* sp. Arz-ArchMG-1 (JQ929040), from Arzakan geothermal spring as a hydrogenotrophic methanogenic archaea sharing 97% 16S rRNA gene sequence similarity with *Methanoculleus* sp. LH2 (DQ987521).

A purple nonsulfur thermotolerant (T_max_ 45 °C) bacterial strain, *Rhodopseudomonas palustris* D-6, was isolated from water/sediment samples of Jermuk hot spring [49,50]. This strain could use organic carbon and nitrogen sources. *R. palustris* D-6 is an active producer of aspartase and acylase. Several aspartase, β-carotene, aminoacylase, glucose isomerase, and inulinase-producing mesophilic phototrophic bacterial strains (*Rhodobacter Rhodopseudomonas*, *Thiospirillum*, and *Thiocapsa*) have also been isolated from the studied geothermal springs. Some of these isolates are good producers of proteins, carbohydrates, and vitamins [49]. The thermophilic bacteria isolated from the Armenian geothermal springs are listed in Table 2.

Culture-dependent studies are important to determine the taxonomic diversity of the springs as well as to understand the ecological role of microbes. Thus, bioprospecting of hot spring microbes has also been considered to investigate the use of these resources for commercial applications. Most studies have focused on important enzymes such as lipases, proteases, and amylases. Thermophilic microbes capable of producing hydrolases have also been reported in geothermal springs worldwide. The distinctive feature of the majority of hydrolase-producing bacilli isolated from Armenian hot springs is their ability to excrete an assortment of extracellular enzymes. Thus, *Anoxybacillus rupiensis* Arzakan-2 and *Geobacillus*
*stearothermophilis* H-2 were efficient producers of all three types of thermostable enzymes, combined almost in an equal ratio, whereas *P. toebii* Tatev-5 and *Anoxybacillus* sp. KC-3 were producers of two types of hydrolases [35].

Thermal ecosystems are proving to be an attractive source of EPS-producing thermophiles. EPS produced by thermophilic bacilli are of interest for the study of their biological role and their potential applications in biotechnology. The strains *G. thermodenitrificans* ArzA-6 and *P. toebii* ArzA-8 are among the limited number of reported thermophilic EPS producers. These strains exceed other known thermophilic geobacilli producers because of the high level of polymer synthesis. The ability to produce EPS using fructose as a carbon source, the high T_opt_ for EPS synthesis, and the high content of uronic acids are also unique characteristics of these isolates [41].

The *Ts*K1 DNA polymerase obtained from the thermophile *T. scotoductus* strain K1 also has the potential to be commercialized. The better base insertion fidelity of *Ts*K1 is a feature that demonstrates an advantage over Taq DNA polymerase [43] and can be used in various high-temperature polymerization reactions.

*Treponema* sp. J25, and *Anoxybacillus* sp. K1 strains were subjected to whole-genome shotgun sequencing [37,38]. The whole-genome shotgun project of strain J25 has been deposited at DDBJ/EMBL/GenBank under the accession number PTQW00000000, and that of strain K1 has been deposited in GenBank under the accession number MQAD00000000. Similarly, the draft genome sequence of *T. scotoductus* K1 was reported following its isolation from Karvachar spring [42], and has been deposited in the DDBJ/EMBL/GenBank database under RefSeq assembly accession no. GCF_001294665.1. The distinctive genome features of these strains are given in Table 3.

Comparative genomic analysis of *Treponema* sp. J25 and its closest species, *T. caldarium*, indicated that both thermophilic *Treponema* species have a metabolic capacity for reductive acetogenesis. Unlike *T. caldarium,* the J25 genome possesses homologs of molybdenum-dependent nitrogenase (Mo-nitrogenase). Genomic analyses further confirmed that *Treponema* sp. J25 is the only known thermophilic free-living treponeme with metabolic potential for nitrogen fixation [38].

Gene prediction was performed using the RAST server (http://rast.nmpdr.org/rast.cgi) (accessed on 25 December 2013) and the Genome Annotation Pipeline of NCBI; in total, 2529 genes, including three sets of rRNA genes, were identified in the genome of *T. scotoductus* K1. Similar to *Thermus* spp., all the rRNA genes found were unlinked and located in separate operons. Unlike other strains of *T. scotoductus*, two CRISPR arrays have been identified for *T. scotoductus* K1 [42].

The draft genome of the *Anoxybacillus* sp. K1 contained 2689 predicted coding genes, 115 pseudogenes, and two CRISPR arrays. Comparative genomic analysis of strain K1^T^ and its closest species, *Anoxybacillus flavithermus* DSM2641^T^, revealed a large number of scattered small non-homologous regions (unpublished data).

### 3.2. Cultivation-Independent Studies

To date, only a small fraction of prokaryotes have been isolated and characterized. The application of several molecular biological approaches has elucidated the diversity of microbial communities in different environments. Here, we summarize the available data obtained through molecular and culture-based methods on the diverse prokaryotic communities thriving in Armenian geothermal springs. The metagenomic analysis workflow used to study geothermal springs in Armenia and Nagorno-Karabakh is shown in Figure 2.

The predominance of Proteobacteria (Alpha-, Beta-, Gamma-, and Epsilonproteobacteria), Firmicutes, Bacteroidetes, and Cyanobacteria in the Arzakan geothermal spring has been revealed using a 16S rRNA gene clone library [33]. The abundance of species from the bacterial phyla Proteobacteria, Bacteroidetes, and Cyanobacteria in the Arzakan spring microbial profile was confirmed by DGGE analysis [33,54].

Pyrosequencing using the 454 GS FLX platform of Arzakan spring sediment samples also revealed highly diverse microbial communities, dominated by oxygenic photosynthetic aerobes (Cyanobacteria) but also including Proteobacteria, Bacteroidetes, Chloroflexi, and Spirochaeta. In addition to oxygenic cyanobacterial species, purple nonsulfur anoxygenic phototrophic Betaproteobacteria have been identified as primary producers in the food chains of ecosystems. Chemolithotrophs, such as hydrogen- and sulfide-oxidizing Epsilonproteobacteria and methanotrophic Gammaproteobacteria, also support the primary production of geothermal systems. The majority of obtained pyrotags (84%) were assigned to unknown genera, underscoring the presence of new forms in these ecosystems [48].

The same method has also been used to study the bacterial diversity in Jermuk hot springs, and Proteobacteria, Bacteroidetes, and Synergistetes were found to be the most abundant phyla in their pyrotag dataset [48].

Chemolithoautotrophs capable of using sulfur compounds, Fe^2+^, and/or H_2_ as electron donors (representatives of the genera *Thiobacillus, Sulfuricurvum, Siderooxydans*, and *Hydrogenophaga*) were among the dominant components, indicating their important role in the biogeochemical cycling of biogenic elements [55,56,57,58]. Bacteroidetes were diverse, and only a few OTUs were assigned to chemoorganotrophic representatives of the genus *Lutibacter* [59]. Both Bacteroidetes and Synergistetes are major reducers involved in the mineralization processes in Jermuk springs.

The bacterial populations in the Jermuk geothermal spring were also profiled using DGGE, which revealed the presence of Epsilonproteobacteria, Bacteroidetes, Spirochaetes, Ignavibacteriae, and Firmicutes. The DGGE profile further revealed anaerobic or facultatively anaerobic chemorganoheterotrophic fermentative microorganisms actively involved in the C cycle, as well as H_2_-utilizing and thiosulfate/sulfur-reducing chemolithotrophic bacteria [35].

Archaeal pyrotags affiliated with species of the orders *Methanomicrobiales* and *Methanosarcinales* and relatives of *Methanomassiliicoccus luminyensis* have been detected in Jermuk. Ammonia-oxidizing archaea (AOA) actively involved in the N-cycle, and considered to be the main drivers of ammonia oxidation in these habitats, have been found among the generated pyrotags. The studied hot springs were also populated by as yet uncultivated Crenarchaeota representatives, including species from Miscellaneous and Deep Hydrothermal Vent Group 1 [60].

The representatives of the orders *Methanosarcinales* (*Methanosaeta*, *Methanomethylovorans*, and *Methanospirillum hungatei*) and *Methanomicrobiales* (including species from the genus *Methanoregula*) were inferred to be abundant in both the Arzakan and Jermuk pyrotag datasets. The presence of methanogens, which can use H_2_/CO_2_ and/or formate at 50–55 °C, was unusual because they are reported to grow in moderate- and low-temperature environments. However, obligate acetoclastic species of the genus *Methanosaeta* have been reported to grow at temperatures up to 60 °C in Jermuk [60].

Recently, total DNA isolated from water/sediment samples of Jermuk spring were analyzed using paired-end shotgun sequencing with an Illumina HiSeq2500, which indicated the dominance of Proteobacteria, Firmicutes, and Bacteroidetes. The candidate division WS6 and the candidate phylum Ignavibacteria were also found among the obtained phylotypes. Euryarchaeota, Crenarchaeota, and Thaumarchaeota were the dominant groups in this archaeal community [38].

In the Zuar geothermal spring, bacterial clone library construction based on 16S rRNA genes indicated the presence of representatives originating from the phyla Proteobacteria (42.3%), Firmicutes (19.2%), Bacteroidetes (15.4%), Cyanobacteria (3.8%), Tenericutes (3.8%), and as yet unclassified phylotypes (15.4%) [53].

Metagenome analysis using the Illumina HiSeq platform yielded more than 11 million high-quality sequence reads for water and sediment samples from the Karvachar geothermal spring. The number of reads assigned to bacteria was 94.2% and 83.8% in water and sediment samples, respectively. Archaeal sequence reads formed a minority, comprising 0.04% and 0.07% of the total reads in water and sediment samples, respectively. The thermal water was dominated by Proteobacteria (> 85% of the total reads), followed by Bacteroidetes, Actinobacteria, Firmicutes, Chloroflexi, Ignavibacteriae, and Deinococcus–Thermus. In the prokaryotic population, Cyanobacteria, Nitrospirae, Synergistetes, Acidobacteria, Planctomysetes, Deferribacteria, Spirochaetes, Chlorobi, and Verrcomicrobia formed minorities. Representatives of the *Parvibaculum*, *Pseudomonas*, *Acidovorax*, *Ramlibacter*, *Flavobacterium*, *Rubrivivax*, *Rhodobacter*, *Thiobacillus*, *Melioribacter*, *Shewanella*, and *Chloroflexus* genera were predominant (Figure 3). The relationship of most of the obtained sequences with uncultivated organisms indicates a unique community structure in the studied spring [61]. The dominant bacterial and archaeal phyla obtained with different culture-independent methods are presented in Table 4.

## 4. Correlation between the Geophysiology and Microbiology of Hot Springs in the Lesser Caucasus

Generally, temperature is a determining factor that determines the taxonomic and functional structure of bacteria and archaea in hot springs [62]; the higher the environmental temperature, the lower the microbial diversity. A similar negative correlation has been established for geothermal springs in previous studies [62,63]. In high-temperature hot springs with temperatures greater than 75 °C, thermophiles or hyperthermophiles are basic colonizers and key leaders of biogeochemical cycles [12]. However, in terrestrial thermal springs, with temperatures less than 75 °C, moderately thermophilic and mesophilic phototrophs from the phyla Cyanobacteria, Chloroflexi, and Proteobacteria are the major primary producers [64]. Archaeal phyla such as Crenarchaeota, Euryarchaeota, and Thaumarchaeota are also commonly detected in geothermal systems [65,66].

Much higher temperatures at deeper levels (in some cases up to 99 °C) have been determined by geophysicists in Armenian thermal springs [28]. This is probably due to the geotectonic and volcanic activity of this region. Therefore, the detection of resident thermophilic microbes in mesothermal springs is not surprising.

The temperature value of most Armenian hot springs at the outlet is below 50 °C. These mesothermal springs harbor Proteobacteria, Bacteroidetes, Cyanobacteria, and Firmicutes. Higher temperature springs are occupied by thermophiles belonging to the phyla Actinobacteria, Deinococcus–Thermus, and Aquificae. The most versatile phylum reported in both warm and hot springs was Firmicutes. These observations are in accordance with many global studies that indicate the dominance of Aquificae, Deinococcus–Thermus, and Firmicutes in hot springs and the prevalence of Cyanobacteria, Chloroflexi, and Proteobacteria in mesothermal springs [62]. The neutral pH and moderate temperature of geothermal systems in the studied Armenian springs permit phototrophy. Due to the obvious light effects at the spring outlet, phototrophic bacteria are often colonized as visible, microbial growth-forming, laminated mats or streamers.

In the studied springs, archaeal representatives formed a minority of the prokaryotes found. Earlier Archaea never dominated mesothermal and circumneutral environments [14,62]. Considering the ability of thermophilic Archaea to grow under extreme thermal conditions, it seems that high-temperature environments should be mainly populated by Archaea [20,67,68]. Surprisingly, studies on hot spring microbiota have revealed the predominance of bacteria in such environments [11,69]. However, this phenomenon is not well understood.

Neutral pH was the optimal level for the growth of almost all bacterial isolates, and they did not grow under moderate or strictly acidic conditions. The dominance of isolates belonging to Firmicutes and Proteobacteria in the Armenian geothermal springs is in line with microbial assemblages distributed in hot springs with pH ≥ 7 globally [62,70].

The set of physicochemical and edaphic conditions along with the nutritional status in a natural habitat may drive the development of a particular microbial group or population.

Moreover, environmental factors can allow the natural selection of a few species capable of dominating and multiplying in ecologically relevant niches. Limited carbon and nitrogen sources, high mineralization, and the high temperature of the studied springs also promoted the development of a unique population dominated by several thermophilic bacilli, including *Geobacillus*, *Parageobacillus*, and *Anoxybacillus* spp. Bacilli are the most versatile organisms and, being eurytherms, they can function at a wide range of temperatures [35]. As they are chemoorganotrophic aerobes or facultative aerobes, thermophilic bacilli actively contribute to the biogeochemical cycles of biogenic elements under extreme temperature conditions. The abundance and diversity of *Bacillus* species in hot springs are in accordance with the observations reported internationally, indicating a similar pattern [71,72,73].

Dissolved gases (H_2_, CO_2_, H_2_S, and CH_4_) are possible limiting factors for microbial diversity. The geothermal systems of the Lesser Caucasus are known to be highly mineralized and have a strong influence on microbial community composition. Recent studies have highlighted the strong influence of biogeography and geological history on the structure of autochthonous microbiota in geothermal springs [74,75]. This was confirmed by the close resemblance of microbial diversities in springs located along the Alpide orogenic belt. Thus, it is not surprising to find a close relationship between thermophilic bacilli isolated from Armenian hot springs and those isolated from high-altitude geothermal springs in the Indian and Nepalian Himalayas [72,76,77], the Aegean region and Anatolian plateau in Turkey [73,78], and the Rhodope Mountains in Bulgaria [71]. Presumably, ongoing active seismic processes affect the biodiversity along the belt, indicating the occurrence of biogeographical structuring.

The microorganisms detected in the studied geothermal springs are phylogenetically diverse and tend to be phenotypically associated with a fermentative, photosynthetic, aerobic, and anaerobic chemoorganotrophic and chemolithorophic respiratory metabolism.

The microbial communities of the studied hot springs are likely able to utilize biopolymers (from fallen plants) and other natural organic matter as carbon sources. The presence of thermophilic bacilli with high hydrolytic activity is an indicator of active degradation of natural biopolymers [35].

The hypothesis regarding autotrophic carbon fixation is supported by the identified presence of phototrophic Cyanobacteria and Proteobacteria, as well as green sulfur bacteria (Chlorobi), chemolithotrophic *Nitrospira* spp. and representatives from the phylum Proteobacteria [33,47,48]. Chemolithotrophy is important for primary productivity. The Proteobacteria found in the studied geothermal springs were related to obligate or facultative chemolithoautotrophs and were capable of oxidizing sulfur compounds, Fe^2^^+^, and/or H_2_ as electron donors. Methanogenic Euryarchaeota are also involved in carbon fixation via the reductive acetyl CoA pathway [33,48].

Other types of methanogenic pathways utilizing methanol, acetate, and methylamine are also known to be involved in methane production [48]. Such methanogenic thermophiles were confirmed to be present in the hot springs of Arzakan and Jermuk. Gammaproteobacterial methanotrophs belonging to *Methylocaldum* spp. were found in the Akhurik and Arzakan geothermal springs, indicating the importance of methane in C cycling [33,45].

The presence of anaerobic Firmicutes and Proteobacteria, as well as *Methanosaeta* spp., indicates the potential of the studied spring community to fix atmospheric nitrogen. The thermophilic representatives of the genus *Treponema* in the Jermuk spring with a metabolic potential for nitrogen fixation might play a key role in N cycling in this geothermal niche [38].

Low oxygen content provides evidence of the presence of denitrifying bacteria that can use nitrate or nitrite as terminal electron acceptors in anaerobic respiration by reducing them to molecular nitrogen.

The detection of ammonia-oxidizing archaea (“Candidatus Nitrososphaera gargensis” and the not yet cultivated Thaumarchaeota) and bacterial nitrite-oxidizing bacteria, such as *N. calida* and *N. moscoviensis*, in the Jermuk spring indicates their key role in the nitrification processes in these habitats [47,48].

Sulfur metabolism involves sulfur oxidation and sulfur reduction. Both metagenome data and positive enrichment of sulfate reducers related to the genus *Desulfomicrobium* confirmed the generation of reductive forms of sulfur compounds in the Jermuk hot spring [38]. The generation of hydrogen sulfide is also an important transformation in sulfur metabolism for the synthesis of sulfur-containing amino acids, and it can also be used as an electron donor by sulfur-oxidizing chemolitotrophs.

The results of phylogenetic analysis obtained from Karvachar spring were compared with the results reported for similar environments located within the Alpide belt (Table 5). Investigation of samples from seven terrestrial hot springs from Turkey, Bulgaria, and India revealed that the number of dominant or major phyla varied from four to seven. Despite phylogenetic diversity, distinct phylogenetic groups (such as Proteobacteria, Bacteroidetes, and Firmicutes) were repeatedly detected in geothermal springs. Sequences assigned to Proteobacteria were found in all the compared springs. However, different Proteobacteria groups were established in different springs. Deinococcus–Thermus was established in four of the springs (Karvachar, Bulgarian, and two Indian hot springs). This genus appears to be present in terrestrial hot springs. Cyanobacteria have also been established as an important part of community metabolism in many continental hot springs [79,80,81,82,83,84,85].

## 5. Conclusions

Elucidation of the structural and functional composition of the microbiota in hot springs is important to understand the biogeochemical cycles of elements and to reveal the biotechnological potency of the microbes harbored in these springs. In this review, we summarized the results of investigations on microbial communities thriving in nine geothermal springs of the Armenian Highland. Culture-independent analysis revealed that the key and dominant bacterial phyla in all studied springs were Proteobacteria, Bacteroidetes, Cyanobacteria, and Firmicutes. The representatives of dominant bacterial phyla are actively involved in the biogeochemical cycling of carbon, nitrogen, and sulfur in thermal ecosystems. Cyanobacteria are the main producers of the ecological chain, along with chemolithotrophic producers. Archaeal prokaryotes form a minority of the microbial community (comprising < 1% of the detected microbes). Archaeal populations have been presented by both cultivable and as yet uncultured representatives of the phyla Euryarchaeota, Crenarchaeota, and Thaumarchaeota.

Temperature was established as a significant environmental factor determining the microbiota of the studied geothermal springs. Biogeography and geological history, along with abiotic factors such as temperature, pH, and mineralization, collectively contribute to the dynamics and structure of microbial populations. Many new thermophilic bacterial and archaeal strains have been isolated from these springs and evaluated for their biotechnological potential. Strains capable of producing thermostable hydrolases (proteases, lipases, and amylases), EPS, DNA polymerases, and other bioactive compounds have been identified and deposited. Such complex studies and detailed characterization of thermophilic microbes from Armenian hot springs confirm the potential of the studied springs as a source of new and biotechnologically prospective thermophiles.

## Figures and Tables

**Figure 1 microorganisms-09-01473-f001:**
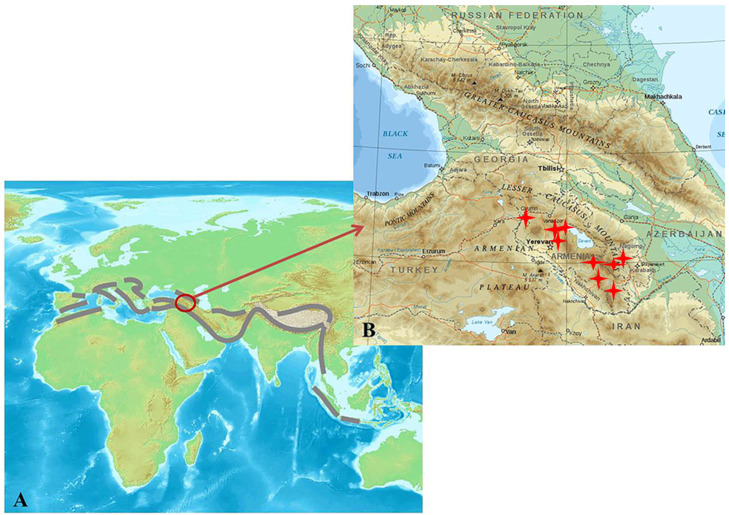
Alpide (Alpine–Himalayan) orogenic belt (marked by brown intermittent lines) (**A**) and the Greater and Lesser Caucasus Mountains (**B**). The locations of the studied geothermal springs are indicated by red four-point stars. The maps were sourced from: https://en.wikipedia.org/wiki/Alpide_belt and https://en.wikipedia.org/wiki/Caucasus (accessed on 2 March 2008 and 23 July 2016, respectively).

**Figure 2 microorganisms-09-01473-f002:**
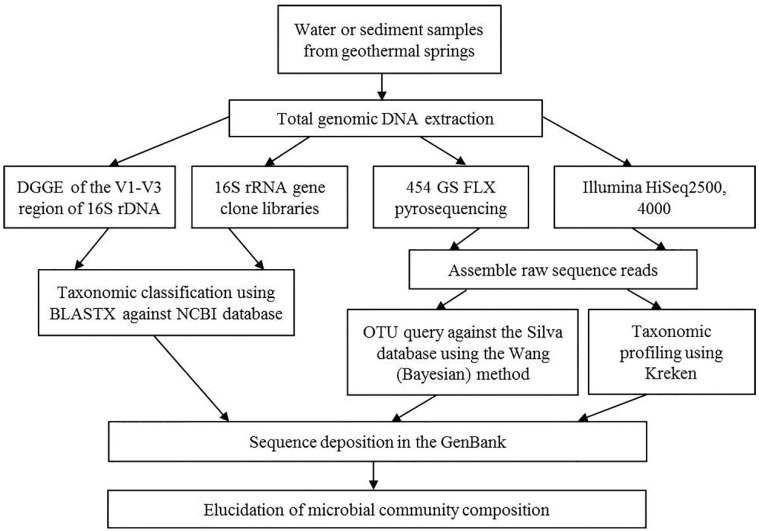
Culture-independent analysis workflow.

**Figure 3 microorganisms-09-01473-f003:**
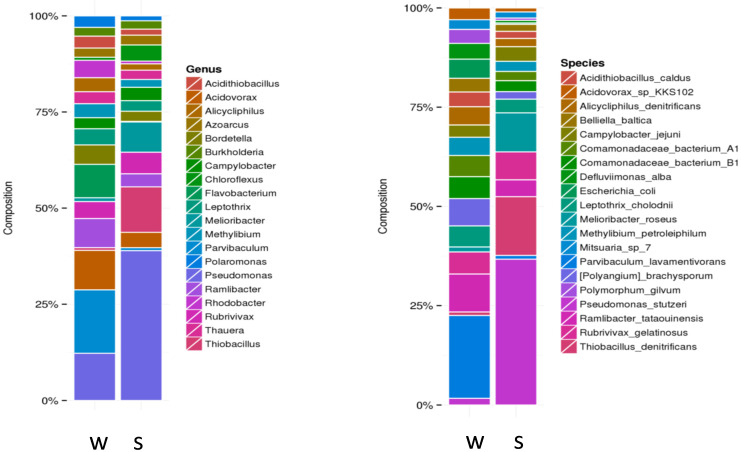
Genus- and species-level distributions in water (w) and sediment (s) samples from Karvachar geothermal spring.

**Table 1 microorganisms-09-01473-t001:** General characteristics of Armenian and Nagorno-Karabakhian geothermal springs.

Thermal Mineral Spring	Spring Location	Altitude, m Above Sea Level	pH	Conductivity, μS/cm	Temperature of Water in the Outlet, T, °C	Description	Photographs
Armenia							
Akhurik	40°44′34.04″ N 43°46′53.95″ E	1490	6.5	2490	30	Hydrocarbonate-sulphate sodium-magnesium type of spring. Slightly degassing. Sand at the bottom.	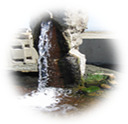
Arzakan	40°27′36.10″ N 44°36′17.76″ E	1490	7.2	4378.3	44	Hydrocarbonate sodium class of mineral spring with a high concentration of dissolved minerals (of which > 20% is HCO^3−^ and > 20% is Na^+^). Slightly degassing. Silicate sand at the bottom.	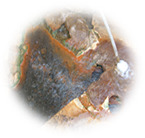
Bjni	40°45’94.44” N 44°64’86.11” E	1610	6.2–7.0	4138.3	30–37	Chloride-hydrocarbonate sodium type of spring. Sand at the bottom.	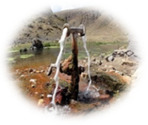
Hankavan	40°63′26.50″ N 44°48′46.00″ E	1900	7.0–7.2	6722.9	42–44	Hydrocarbonate-chloride sodium spring. Vigorously degassing. Silicate sand at the bottom.	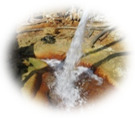
Jermuk	39°96′63.90″ N 45°68′52.80″ E	2080	7.5	4340	> 53	Carbon hydro-sulphate-sodium water source. Sand at the bottom.	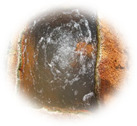
Tatev	39°23′76.00″ N 46°15′48.00″ E	960	6.0	1920	27.5	Carbon-bicarbonate calcium water sources. Many bubbling sources and no visible outflow. Clays and sands at the bottom. Source was left in its natural form; no trace of human intervention was found.	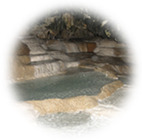
Uytc (Uz)	39°31′00″ N 46°03′09″ E	1600	6.23	2700	25.8	Hydrocarbonate-chloride-sulphate sodium source. Sand at the bottom.	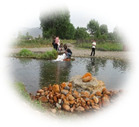
Nagorno-Karabakh							
Karvachar	40°17′41.00″ N 46°27′50.00″ E	1584	7.3	4600	70	Hydrocarbonate-sulphate sodium source. Clear water and fine clay at the bottom.	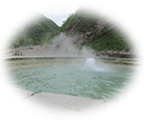
Zuar	40°02′47.60″ N 46°14′09.30″ E	1520	7.0	4300	42	Hydrocarbonate-sulphate sodium source. Clay and sand at the bottom.	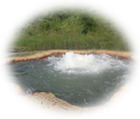

**Table 2 microorganisms-09-01473-t002:** Summary of the thermophilic bacteria isolated from Armenian geothermal springs.

Geothermal Spring	Bacterial and Archaeal Genera	Species	References
Armenia			
Akhurik	*Bacillus, Brevibacillus,* *Thermoactinomyces, Rhodobacter, Thiospirillum,* *Methylocaldum*	*B. licheniformis, B. pumilus, B. murimartini, B. psychrosaccharolyticus, B. borstelensis, Thermoactinomyces* sp., *R. sulfidophilus, T. jenense, Methylocaldum* sp.	[35,39,44,45,49]
Arzakan	*Bacillus* *, Paenibacillus,* *Geobacillus, Parageobacillus, Anoxybacillus, Rhodobacter,* *Rhodopseudomonas, Thiocapsa, Nitrospira, Arcobacter, Methanoculleus*	*B. licheniformis, B. simplex, Paenibacillus* sp., *G. thermodenitrificans, G. stearothermophilus, P. toebii, P. caldoxylosilyticus, A. rupiensis*, *R. shpaeroides, R. palustris, T. roseopersicina, N. calida, N. moscoviensis, Arcobacter* sp., *Methanoculleus* sp.	[33,35,36,37,49]
Bjni	*Bacillus*, *Ureibacillus, Parageobacillus, Anoxybacillus, Rhodobacter* *Rhodopseudomonas, Thiocapsa*	*B. licheniformis, U. thermosphaericus, P. toebii, Anoxybacillus* sp., *R. shpaeroides, R. palustris, T. roseopersicina*	[35,49]
Hankavan	*Bacillus, Brevibacillus, Geobacillus, Parageobacillus, Anoxybacillus*	*B. licheniformis,**B. thermoruber, G. stearothermophilus, P. toebii, Anoxybacillus* sp.	[35]
Jermuk	*Bacillus* *, Parageobacillus, Geobacillus, Anoxybacillus, Desulfomicrobium, Desulfovibrio, Treponema, Rhodobacter, Rhodopseudomonas, Thiospirillum, Nitrospira*	*B. licheniformis, P. caldoxylosilyticus, Geobacillus* sp., *A. gonensis, A. kestanbolensis, A. flavithermus, D. thermophilum, D. psychrotolerans, Treponema* sp., *R. capsulatus, R. shpaeroides, R. palustris, T. jenense, N. calida, N. moscoviensis*	[35,38,47,49]
Tatev	*Bacillus, Geobacillus, Anoxybacillus*, *Parageobacillus, Thermoactinomyces*	*B. aerius, Geobacillus* sp., *Anoxybacillus* sp., *P. toebii, T. vulgaris*	[35,44]
Uyts	*Bacillus, Ureibacillus, Aeribacillus, Anoxybacillus, Geobacillus*	*B. licheniformis, B. glycinifermentans, U. terrenus, A. pallidus, Anoxybacillus* sp., *Geobacillus* sp.	[35]
Nagorno-Karabakh			
Karvachar	*Bacillus, Aeribacillus, Anoxybacillus, Thermus*	*B. licheniformis, B. simplex, A. pallidus, A. suryakundensis, A. flavithermus, T. scotoductus*	[35,42,43]
Zuar	*Anoxybacillus, Geobacillus*	*A.**rupiensis*, *Geobacillus* sp.	[35]

**Table 3 microorganisms-09-01473-t003:** Genome features and statistics of strains *Anoxybacillus* sp. K1, *T. scotoductus* K1, and *Treponema* sp. J25.

Genome Features	Strains		
	*Anoxybacillus* sp. K1	*T. scotoductus* K1	*Treponema* sp. J25
GenBank accession	MQAD00000000	LJJR01000000	PTQW00000000
Contigs	48	55	72
Base pairs	2,722,200	2,379,636	3,180,620
GC %	41.6	65.2	49.6
rRNAs (5S, 16S, 23S)	5, 4, 1	3, 1, 3	1, 1, 1
tRNAs	64	48	44
Genes (total)	2883	2529	2744
CRISPR arrays	2	2	3

**Table 4 microorganisms-09-01473-t004:** Bacterial and archaeal phyla detected in culture-independent studies on Armenian geothermal spring microbiomes.

Geothermal Spring	Approach	Dominant Bacterial and Archaeal Phyla	Accession Number	References
Armenia				
Arzakan	Shotgun pyrosequencing of the V4 region on the 454 GS FLX platform	Cyanobacteria, Proteobacteria, Bacteroidetes, Chloroflexi, Spirochaeta, Euryarchaeota, Crenarchaeota (the as yet uncultivated group, MCG)	SRR747863	[48]
	Bacterial 16S rRNA gene library	Bacteroidetes, Cyanobacteria, Betaproteobacteria, Gammaproteobacteria, Epsilonproteobacteria, Firmicutes, Alphaproteobacteria	JQ929026–JQ929037	[33]
	Archaeal 16S rRNA gene library	Euryarchaeota, AOA Thaumarchaeota ‘‘Ca. Nitrososphaera gargensis’’, as yet uncultivated Crenarchaeota (MCG and DHVC1 groups)	KC682067–KC682083	[48]
	DGGE	Beta-, Epsilon-, and Gammaproteobacteria, Bacteroidetes, Cyanobacteria	JX456536–JX456538	[33,34]
Jermuk	Shotgun pyrosequencing of the V4 region on the 454 GS FLX platform	Proteobacteria, Bacteroidetes, Synergistetes Euryarchaeota, as yet uncultivated Crenarchaeota (MCG and DHVC1 groups)	SRR747864	[48]
	Illumina HiSeq2500 paired-end sequencing	Proteobacteria, Firmicutes, Bacteroidetes, candidate division WS6, candidate phylum Ignavibacteria, Euryarchaeota, Crenarchaeota, Thaumarchaeota	-	[38]
	Archaeal 16S rRNA gene library	Euryarchaeota, AOA Thaumarchaeota ‘‘Ca. Nitrososphaera gargensis’’, as yet uncultivated Crenarchaeota (MCG group)	KC682084–KC682097	[48]
	DGGE	Epsilonproteobacteria, Bacteroidetes, Spirochaetes, Ignavibacteriae, Firmicutes		[34]
Nagorno-Karabakh				
Karvachar	Bacterial 16S rRNA gene library	Proteobacteria, Cyanobacteria, Bacteroidetes, Chloroflexi, Verrumicrobia, Planctomycetes	-	[51,52]
	DGGE	Bacteroidetes, Firmicutes	-	[34,51]
	Whole-metagenome shotgun sequencing using the Illumina Hiseq 4000 platform	Actinobacteria; Alpha-, Beta-, Delta-, Epsilon-, and Gammaproteobacteria; Bacteroidetes/Clorobi; Firmicutes; Clamydiaae; Cyanobacteria/Melainabacteria; Fusobacteria; Synergistia	-	[51]
Zuar	Bacterial 16S rRNA gene library	Proteobacteria, Firmicutes, Bacteroidets, Cyanobacteria, Tenericutes, as yet unclassified phylotypes	-	[53]

**Table 5 microorganisms-09-01473-t005:** Comparison of bacterial diversity in different terrestrial hot springs.

Hot Spring	T (°C)/pH	Main Ions in Water	Dominant or Major Bacterial Phyla *	Approach	Reference
Karvachar, Nagorno-Karabakh	70/7.3	*Anions*: HCO^3−^, SO_4_^2−^ *Cations*: Na^+^	Proteobacteria Bacteroidetes Ignavibacteriae Actinobacteria Chloroflexi Deinococcus–Thermus	Illumina HiSeq 4000	[61]
Kaklik, Turkey	35.4/6.77	*Anions*: Cl^−^, SO_4_^2−^ *Cations*: Mg^2+^, Ca^2+^, Fe	Proteobacteria Bacteroidetes Verrucomicrobia Firmicutes Actinobacteria Nitrospirae Acidobacteria	454 pyrosequencing	[79]
Orhaneli, Bursa, Turkey	68/7.8	*Anions*: Cl^−^, NO_3_^−^, PO_4_^3−^ *Cations*: Na^+^, NH_3_-N, K^+^, Mg^2+^, Ca^2+^	Proteobacteria Chloroflexi Bacteroidetes Firmicutes	454 pyrosequencing	[80]
Rupi Basin, Bulgaria	79/8.6	*Anions*: Cl^−^, SO_4_^2−^, HCO^3−^, HS^−^ *Cations*: Na^+^, K^+^, Ca^2+^	Proteobacteria Hydrogenobacter Deinococcus–Thermus Cyanobacteria Thermotoga Cytophaga	Clone library	[81]
Polichnitos, Greece	80/7.5	*Anions*: Cl^−^, SO_4_^2−^, NO_3_^−^ *Cations*: Na^+^, K^+^, Mg^2+^, Ca^2+^, NH_4_^+^	Proteobacteria Cyanobacteria Firmicutes	Clone library	[82]
Polok, Sikkim Himalaya, India	62/8.0	Nd	Proteobacteria Firmicutes Chloroflexi Deinococcus–Thermus Aquificae Bacteroidetes	Illumina MiSeq 2500	[83]
Unkeshwar, Maharashtra, India	50–60/7.3	*Anions*: PO_4_^3-^, SO_4_^2-^	Actinobacteria Verrucomicrobia Bacteriodes Deinococcus–Thermus Firmicutes	Illumina HiSeq 2500	[84]
Jakrem, Meghalaya, India	46/9.0–10.0	*Anions*: SO_4_^2−^, NO_3_^−^, Cl^−^ *Cations*: Mg^2+^, Na^+^, K^+^, Ca^2+^, Fe	Firmicutes Chloroflexi Proteobacteria	Illumina MiSeq	[85]

* Phylogenetic groups representing more than 2% of the community sequences are presented. Nd—not described.

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
