# Peer review of "Microbial Diversity of Terrestrial Geothermal Springs in Armenia and Nagorno-Karabakh: A Review"

_microorganisms, 2021, doi:10.3390/microorganisms9071473_

Round 1
Reviewer 1 Report
This review manuscript describes on the microbiological aspects on terrestrial geothermal springs of Armenia and Nagoro Karabakh.
Major:
- “3.1. Cultivation-dependent studies” have endless descriptions of the properties of the isolates. It is recommended that the authors focus on the isolates that are quite distinctive and explain its abundance ratio in the next-generation sequence analysis, including the ecological functions in the field.
- Please describe distinctive feature on the results of the whole genome sequences of the isolates.
- The results of the next-generation sequence analysis are also described in simply, but what the reader wants to know is the comparison of data between geothermal springs, which is the theme this time, and the comparison with data from other similar environments in totally different places in the world.
- Table 1: Contents of the table was absent.
- Figure 3: While there are many types of legends, there are few types of colors in actual bar graphs, which is difficult to understand.
Minor:
- Title: Aremenia → Armenia
- L39: Please insert references.
- L111: Please insert references.
Reviewer 2 Report
The review article by Ani Saghatelyan and co-workers describe The Microbiological diversity of terrestrial geothermal springs of Armenia and Nagorno Karabakh. Although it is an interesting piece of work, the manuscript needs further more revisions.
Abstract: Authors may improve this part, and please be precise
Introduction: It still needs more inputs and extensive modification.
Figure 1: Modify it and give the appropriate legend
3.1: There is no table heading
Line no 112-115: references missing
For cultivation dependent approach, authors should give a clear table for a better understanding of readers.
Figure 2: it should be improved
DGGE: please provide the entire form for the first time in the text
The conclusion must be enhanced
Round 2
Reviewer 1 Report
The review manuscript described on the microbiological diversity of terrestrial geothermal springs of Armenia and Nagorno Karabakh has been revised.
This time, it is premised on the analysis of the microbial flora and the isolation of bacteria from special environments that is unique in the world, so if possible, particular points of the result attributed to the unique environment should be described in “Abstract”.
L139-140: The authors described that the most abundant species were Bacillus licheniformis, Parageobacillus toebii and Anoxybacillus flavithermus. Are they exhibited 100% identical to known species in 16S rRNA gene sequence?
Description of cultivation-dependent studies is not exciting. Please focus on the particular things. It should be described what the author can say comparisons with other studies. What can be said in total glance of view through the investigations.
Whole genome analyses: Please present particular characteristics compared with isolates from other environments.
Throughout this manuscript, it was made by the list of information, there is no deep insight of the authors from a bird's-eye view of various research examples. This reviewer think that the authors should mention more about the relationship between the environmental factors in the field and the existing microorganisms.
Author Response
Reviewer 1
- The review manuscript described on the microbiological diversity of terrestrial geothermal springs of Armenia and Nagorno Karabakh has been revised. This time, it is premised on the analysis of the microbial flora and the isolation of bacteria from special environments that is unique in the world, so if possible, particular points of the result attributed to the unique environment should be described in “Abstract”.
RESPONSE:
Thank you for comment and suggestion.
According to the comments we have revised and added information describing uniqueness of the environment. Please find revised abstract in main text as well as below:
Abstract: In recent years, the microbial diversity of high-altitude geothermal springs have been assessed to explore their biotechnological potency. Despite intensive microbiological studies of geothermal springs worldwide, very little is known about microbiota of similar ecosystems located on Armenian Highland. As of today, the microbiota of nine high-altitude mineralized geothermal springs (of temperatures ranging from 25.8 to 70°C and pH 6.0-7.5) in Armenia and Nagorno Karabakh have been investigated. All geothermal springs are high elevated altitudes ranging from 960 to 2090 m above sea level located on the Alpide (Alpine–Himalayan) orogenic belt, a seismically active region in the world. The mixed-cation mixed-anion compositions with total mineralization is 0.5 mg/l, have been found for all studied thermal springs. Studies have been performed to determine the taxonomic diversity hot spring microbiomes by culture-independent approaches including Denaturing Gradient Gel Electrophoresis (DGGE), 16S rRNA gene library construction, 454 pyrosequencing and Illumina Hiseq. Bacterial phyla Proteobacteria, Bacteroidetes, Cyanobacteria and Firmicutes, were the predominant life forms in the studied springs. Archaea mainly representatives of phyla Euryarchaeota, Crenarchaeota and Thaumarchaeota appeared to be a minority, composing less than 1 % of the prokaryotic community. Comparison of microbial diversity in the spring from Karvachar with that described from other terrestrial hot springs revealed that Proteobacteria, Bacteroidetes, Actinobacteria and Deinococcus-Thermus are common basic bacterial groups for terrestrial hot springs. Contemporaneously, specific bacterial and archaeal taxons were observed in different springs. Evaluation of carbon, sulfur, and nitrogen metabolism hot springs community has been revealed diversity in terms of metabolic activity. Temperature was important factor for shaping of the springs’ microbial communities. Overall, the diversity, as well as the richness of microbiota were negatively affected by increasing temperature. Other abiotic factors including pH and mineralization, as well as geological history have impact on structure and function of microbial community. More than 130 bacterial and archaeal strains (Bacillus, Geobacillus, Parageobacillus, Anoxybacillus, Paenibacillus, Brevibacillus Aeribacillus, Ureibacillus, Thermoactinomyces, Sporosarcina, Thermus, Rhodobacter, Thiospirillum, Thiocapsa, Rhodopseudomonas, Methylocaldum, Desulfomicrobium, Desulfovibrio, Treponema, Arcobacter, Nitropspira and Methanoculleus) some of which are potentially representatives of novel species (sharing 91-97% sequence identity with their closest match in GenBank) and producers of thermozymes and biomolecules with potential biotechnological applications, have been reported. Whole genome shotgun sequencing of T. scotoductus K1, as well as of potentially new species Treponema sp. J25 and Anoxybacillus sp. K1 were performed. Most of the phyla identified by 16S rRNA were also found using the metagenome approaches.
- L139-140: The authors described that the most abundant species were Bacillus licheniformis, Parageobacillus toebii and Anoxybacillus flavithermus. Are they exhibited 100% identical to known species in 16S rRNA gene sequence?
RESPONSE:
Thank you for comment. We totally agree with reviewer. It is not easy to argue that Bacillus licheniformis, Parageobacillus toebii and Anoxybacillus flavithermus are most abundant species because they didn’t exhibit 100% identity to known species. We revised that part of manuscript by indicating abundance of representatives of genera Bacillus, Geobacillus/Parageobacillus and Anoxybacillus.
Revised version of the sentence is following:
‘’Representatives of genera Bacillus, Geobacillus/Parageobacillus and Anoxybacillus were most abundant.’’
- Description of cultivation-dependent studies is not exciting. Please focus on the particular things. It should be described what the author can say comparisons with other studies. What can be said in total glance of view through the investigations.
RESPONSE:
Thank you for remark. We added special section to describe importance of culture-dependent investigations: We tried to show also some advantages or biotechnological potential of isolates compared with isolates obtained from other similar environments. Please find revised version of manuscript as well as below:
‘’Results of culture-dependent studies are important not only to find out taxonomic diversity of springs, but also to understand ecological role of microbes. Attention was also paid to bioprospecting of hot springs microbes with a view to use these resources for commercial application.
Majority of the studies were focused on important enzymes like lipase, protease and amylase. Thermophilic microbes able to produce hydrolases have also been reported in geothermal springs worldwide. Distinctive feature of majority of hydrolase-producing bacilli isolated from Armenian hot springs were its ability to excrete an assortment of extracellular enzymes. Thus, A. rupiensis Arzakan-2 and G. stearothermophilis H-2 were efficient producers of all three types of thermostable enzymes in combination almost in equal ratio, while P. toebii Tatev-5 and Anoxybacillus sp.KC-3 were producers of two types of hydrolyses [35].
The thermal ecosystems are proving to be source of EPS producing thermophiles. EPS produced by thermophilic bacilli are the object of interest to study their biological role and the potential applications in biotechnology. The strains G. thermodenitrificans ArzA-6 and G. toebii ArzA-8 are among the limited number of reported thermophilic EPS producers. The strains exceed other known thermophilic geobacilli producers in light of the high level of polymer synthesis. The ability to produce EPS using fructose as a carbon source, as well as the high Topt for EPS synthesis, and the high content of uronic acids are also unique characteristics for isolates [41].
The TsK1 DNA polymerase obtained from thermophile T. scotoductus strain K1 has also potential to be commercialized. The better base insertion fidelity of TsK1 is shown to be distinguishable feature demonstrating advantage over the Taq DNA polymerase [43]. It is suggested to use in various high-temperature polymerization reactions.’’
- Whole genome analyses: Please present particular characteristics compared with isolates from other environments.
RESPONSE:
Thank you for comment. We added some information regarding to particular characteristics of isolates compared with isolates from other environments. Please find added information in revised manuscript, as well as below:
‘’The comparative genomic analysis of Treponema sp. J25 and its closest species T. caldarium indicated that both thermophilic treponemas have a metabolic capacity of reductive acetogenesis. Unlike T. caldarium, the J25 genome possesses homologues of molybdenum-dependent nitrogenase (Mo-nitrogenase). The genomic analyses confirmed that Treponema sp. J25 is the only known thermophilic free-living treponeme with a metabolic potential for nitrogen fixation [38].
Gene prediction analyzed using by RAST server (http://rast.nmpdr.org/rast.cgi) and Genome Annotation Pipeline of NCBI, identified a total of 2,529 genes, including 3 sets of rRNA genes for genome of T. scotoductus K1. Similar to Thermus spp., all found rRNA genes are unlinked and located in separate operons. Unlike other strains of T. scotoductus, two CRISPR arrays were identified for T. scotoductus K1 [42].
The draft genome of Anoxybacillus sp. K1 contained 2689 predicted coding genes, 115 pseudogenes, and two CRISPR arrays. The comparative genomic analysis of strain K1T and its closest species Anoxybacillus flavithermus DSM2641T revealed a large number of scattered small non-homologous regions (unpublished data).’’
- Throughout this manuscript, it was made by the list of information, there is no deep insight of the authors from a bird's-eye view of various research examples. This reviewer think that the authors should mention more about the relationship between the environmental factors in the field and the existing microorganisms.
RESPONSE:
Thank you for remark here. We have a separate section (Correlation between geophysiology and microbiology of the hot springs in the Lesser Caucasus) to discuss the relationship between the environmental factors in the field and the existing microorganisms. We revised that section by adding information also the role of detected microbes in cycling of biogenic elements. Please find added information in added section, as well as below:
‘’Microorganisms detected in studied geothermal springs are phylogenetically diverse and tend to be phenotypically associated with fermentative, photosynthetic, aerobic and anaerobic chemoorganotrophic and chemolithorophic respirative metabolism.
Microbial communities of studied hot springs are likely able to utilize biopolymers (from fallen plants) and other natural organic matters as carbon sources. The presence of thermophilic bacilli with high hydrolytic activity is indicator of active degradation of natural biopolymers [35].
Autotrophic carbon fixation supported by phototrophic Cyanobacteria and Proteobacteria, as well as green sulfur bacteria (Chlorobi), and chemolithotrophic Nitrospira spp. and representatives from phylum Proteobacteria [33,47,48]. Chemolithotrophy is important in primary productivity. Proteobacteria found in studied geothermal springs were related to obligate or facultative chemolithoautotrophs capable to oxidize sulfur compounds, Fe2+ and/or H2 as electron donors. Methanogenic Euryarchaeota also involved in carbon fixation by reductive acetyl-CoA pathway [33,48].
It was shown also other types of methanogenic pathways utilizing methanol, acetate and methylamine for methane production [48]. Such methanogenic thermophiles were confirmed to be present in Arzakan and Jermuk hot springs. Gammaproteobacterial methanotrophs, Methylocaldum spp., found in Akhurik and Arzakan geothermal springs indicated its importance in C cycling [33,45].
The presence of anaerobic Firmicutes and Proteobacteria, as well as Methanosaeta spp. indicates the potential of the studied spring community for fixing atmospheric nitrogen. Thermophilic representatives of genus Treponema in Jermuk spring with a metabolic potential for nitrogen fixation might play a key role in N cycling in its geothermal niche [38].
Low oxygen content supports the presence of denitrifying bacteria able to use nitrate or nitrite as terminal electron acceptors in anaerobic respiration by reducing them up to molecular nitrogen.
The detection of the ammonia-oxidizing archaea (‘‘Candidatus Nitrososphaera gargensis’’and not yet cultivated Thaumarchaeota) and bacterial nitrite-oxidizing like N. calida and N. moscoviensis in Jermuk spring indicates their key role in nitrification processes of these habitats [47,48].
Sulfur metabolism involves sulfur oxidation and sulfur reduction. Both metagenome data and positive enrichment of sulfate reducers related to genus Desulfomicrobium confirmed generations of reductive form of sulfur compounds in Jermuk hot spring [38]. The generation of hydrogensulfide is an important transformation in sulfur metabolism to synthesize of sulfur-containing aminoacids. It can be used also by sulfur oxidizing chemolitotrophs as electron donor.’’
Reviewer 2 Report
The authors have addressed my comments adequately.
